# Novel Immunohistochemical and Morphological Approaches in a Retrospective Study of Post-Mortem Myocarditis

**DOI:** 10.3390/medicina60081312

**Published:** 2024-08-14

**Authors:** Oana Neagu, Violeta Chirică, Lăcrămioara Luca, Maria Bosa, Alina Tița, Mihail Constantin Ceaușu

**Affiliations:** 1Department of Pathology, University of Medicine and Pharmacy Carol Davila, 050474 Bucharest, Romania; 2Emergency Hospital for Children Grigore Alexandrescu, 011743 Bucharest, Romania; 3National Institute of Legal Medicine Mina Minovici, 077160 Bucharest, Romania; 4National Institute of Endocrinology C.I. Parhon, 011863 Bucharest, Romania

**Keywords:** myocarditis, forensic pathology, immunohistochemistry, CD3, CD163, interleukin 6

## Abstract

*Background and Objectives:* This study presents a retrospective analysis of 26 autopsy cases from a single centre, primarily focusing on forensic cases, with a majority of male individuals. *Materials and Methods:* We systematically analysed autopsy reports and cardiac tissue slides using haematoxylin-eosin stain and immunohistochemistry for CD3, CD163, and IL-6. The histological assessment evaluated key variables such as inflammation severity, necrosis, and background changes using a standardised grading system. Quantitative analysis of immunohistochemical markers was performed, calculating the percentage of positively stained cells within the inflammatory infiltrate. *Results:* The average age was 51.6 years, slightly skewed towards older males. The fatalities varied widely, with sudden death and drug abuse being the most common conditions linked to myocarditis findings on histological examination. A strong correlation was found between the severity of inflammation (measured by size within a myocardium section) and the scoring system based on the number of inflammatory foci per section (*p* ≤ 0.001). Most cases showed mild to minimal fibrosis, with some exhibiting moderate to severe fibrosis, arteriosclerosis, and myocyte hypertrophy. The presence of protein CD3 in the inflammatory infiltrate revealed a moderate inverse correlation between the CD3 values and the severity of inflammation and necrosis, and a strong inverse correlation with neutrophil levels. CD3 levels were higher in sudden death cases and lower in cases with numerous inflammatory foci, highlighting the discreet nature of lymphocytic myocarditis. Macrophage presence, assessed using CD163, showed a moderate inverse correlation with neutrophil levels and significant differences between sudden death and non-sudden death cases. Macrophage-rich inflammation was observed in cases with pneumonia/bronchopneumonia-associated lesions. IL-6 expression showed a moderate direct correlation with inflammation severity (*p* = 0.028), severity of necrosis (*p* = 0.005), and the number of inflammatory foci per section (*p* = 0.047). A moderate inverse correlation was found between CD3 and IL-6 expression (*p* = 0.005). *Conclusions*: These findings highlight the need for a unique immunohistochemical approach in forensic cases of myocarditis, differing from guidelines for endomyocardial biopsies due to diverse inflammatory cells. The study suggests exploring inflammatory chemokines within myocarditis foci for their significance in clinical scenarios. Specifically, IL-6, a crucial pro-inflammatory interleukin, correlated significantly with the severity of inflammation and necrosis (*p* < 0.05). This study provides novel and valuable insights into the histopathological and immunological markers of myocarditis in autopsy cases.

## 1. Introduction

Myocarditis represents one of the most challenging diagnoses in cardiovascular pathologies. Inflammatory changes in the cardiac muscle lead to a constellation of symptoms and evolution, mostly depending on the aetiology and the host immune system [1]. However, a definitive diagnosis is seldom accurately assessed, since it requires an invasive approach. Endomyocardial biopsy, unanimously agreed as the gold standard in order to confirm a diagnosis, is performed in thoroughly selected cases, only in expertise centres, due to a high risk of severe complications [2]. Additionally, because of the patchy nature of the histological features, the sensitivity of the tissue specimens varies between 10% and 80%, depending on the aetiology [3].

The diagnostic criteria were first established in 1987 by Thomas Aretz, using the routine stain haematoxylin-eosin. According to the Dallas criteria, a diagnosis of myocarditis is achieved when an inflammatory infiltrate of the myocardium is associated with myocytes necrosis and/or degeneration, in the absence of ischemic lesions or coronary artery disease [4]. In current practice, these features are insufficient and ambiguous, rendering a low sensitivity for several subtypes of myocarditis. A value of >14 leukocytes/mm^2^ with the presence of T lymphocytes > 7 cells/mm^2^ has been proposed as a more reliable cut-off to reach the histological diagnosis of myocarditis [5]. Moreover, molecular biological techniques, such as amplification methods, particularly polymerase chain reaction (PCR) or nested PCR, supplement the detection of viral genomes even for cases with low copies, in small EMB specimens [6]. 

### 1.1. Immunohistochemistry Analysis in Myocarditis

In a meta-analysis published in 2020, an examination of immunohistochemistry tests utilized in endomyocardial biopsies across 61 studies was conducted to ascertain the potential role of this investigation in targeted therapy for myocarditis or other inflammatory cardiomyopathies [7]. The findings indicated that the predominant markers employed were CD3, which highlights T lymphocytes, and CD68, which identifies macrophages. The focus on immunohistochemistry as a supplementary technique for cardiopathies emerged concurrently with the introduction of the Dallas criteria, yet efforts to establish a standardized set of antibodies and a cut-off point have not been successful [8,9]. Only within the past decade has a cut-off point of 7 T lymphocytes been established to confirm the diagnosis of myocarditis, in accordance with the guidelines set forth by the European Society of Cardiology Working Group on Myocardial and Pericardial Diseases [5]. A Japanese study on cardiac inflammation found that myocardial CD3+ T-lymphocyte infiltration is a significant prognostic marker in Dilated Cardiomyopathy (DCM) [10]. The results indicate that a three-tiered risk-stratification model could improve patient categorization (*p* = 0.001), highlighting the need for a quantitative approach in assessing T lymphocytes in myocardial tissue.

In the realm of cardiomyopathies, macrophages embody another pivotal component within the spectrum of inflammatory cells. In myocarditis, irrespective of its underlying causes, macrophages assume a significant role throughout the entire inflammatory continuum, spanning from the initial reaction to the resolution phase. Recent scholarly publications describe two primary macrophage classifications relevant to immune responses: the M1 subtype, representing the classical activation state and involved in innate pathogen response, and the M2 subtype, associated with tissue repair, clearance of apoptotic debris or necrosis, and immune response remodelling [11,12]. In clinical practice, their assessment commonly involves immunohistochemical analyses employing markers such as CD68 or CD163 [13,14]. Initial studies linked CD68 with M1 macrophages and CD163 with M2, but subsequent research does not support these assertions. Due to macrophage phenotypic plasticity, both markers detect macrophages regardless of phenotype [15,16]. CD163 serves as a specific marker for macrophages, whereas CD68 may also detect dendritic cell subsets or granulocytes [17]. Moreover, in a study on 182 endomyocardial biopsies, a high count of CD163-positive infiltrates demonstrated an independent association with a poorer outcome in multivariate Cox regression analysis (hazard ratio 1.77, *p* = 0.004). In addition, multivariate linear regression analysis indicated that the CD163 cell count was an independent determinant of fibrosis (*p* < 0.001) [18]. Recent research indicates the potential clinical utility of evaluating plasma levels of CD163 in cases of fulminant myocarditis. The elevated plasma levels of CD163 observed in these patients may be associated with immune system imbalances and disruptions in systemic inflammatory responses. Additionally, a significant positive correlation was observed between plasma CD163 levels and concentrations of high-sensitivity C-reactive protein (hs-CRP), suggesting a potential link between plasma CD163 and the systemic inflammatory response in patients with fulminant myocarditis [19].

Another protein under scrutiny in cardiac pathology, both within the plasma and as an immunohistochemical marker, is IL-6, a cytokine primarily characterized by its pro-inflammatory properties and multifaceted mechanisms, which contribute to both adaptive and maladaptive responses [20]. IL-6 plays a pivotal role in myocardial remodelling, cardiomyocyte apoptosis, and the diminishment of myocardial contractility, primarily by fostering the recruitment of inflammatory cells in damaged myocardial tissues [21]. In cardiovascular disease, IL-6 is synthesized by various cell types such as macrophages, monocytes, endothelial cells, vascular smooth muscle cells, and fibroblasts [22]. Its pro-inflammatory effects include inhibiting T-cell apoptosis, recruiting inflammatory cells, and suppressing regulatory T-cell differentiation. It is a key mediator of acute phase reactions and correlates with C-reactive protein (CRP) levels, making both IL-6 and CRP useful inflammation biomarkers in clinical settings. Elevated IL-6 levels are linked to atherosclerosis complications and the progression of cardiac insufficiency, suggesting a significant role for IL-6 in heart failure risk assessment [23,24,25].

A recent study found that individuals with elevated IL-6 levels after 5 years had a higher independent risk of cardiovascular disease (CVD) at 10 years compared to those with lower levels. Including IL-6 in risk assessments improved reclassification by 51% over traditional CVD risk factors. These findings suggest further exploration of pharmaceutical interventions targeting IL-6 signalling, especially in patients with high IL-6 levels or heart failure [26,27]. A recent randomized controlled trial found that the anti-IL-6 medication tocilizumab improved the myocardial salvage index by 5.6 percentage points (*p* = 0.04) when administered within 6 h of symptom onset [28]. Moreover, in a murine model of viral-induced myocarditis, tocilizumab significantly reduced myocarditis severity, evidenced by lower cell necrosis, infiltration, and tissue oedema (*p* = 0.005) [29]. In 2021, Amioka et al. found that patients with higher IL-6 serum levels had significantly more IL-6-expressing inflammatory cells in myocardial samples and were more likely to receive catecholamine therapy (*p* = 0.005) [30].

### 1.2. Myocarditis in Clinical Autopsies and Forensic Practice

The majority of confirmed myocarditis cases transpire in the microscopic findings of autopsy studies. As a consequence, forensic pathologists routinely collect samples from the myocardium for histological examination, regardless of the circumstances of death [31]. Autopsy findings serve as the cornerstone in determining the cause of death, forming the primary evaluation that complements both pre- and post-mortem diagnostic examinations. Integrating these findings within a holistic framework is imperative, as it enables forensic pathologists to offer informed and pertinent conclusions, underscoring the pivotal role autopsies play in the diagnostic process [32]. Moreover, the absence of a universally accepted protocol for differentiating between myocarditis as the primary cause of death and myocardial inflammation as an ancillary finding during post-mortem examinations poses a significant challenge [33]. While some studies have explored methodologies for quantifying this disparity, such as assessing inflammatory cell count and myocyte necrosis, numerous limitations impede the reproducibility and reliability of such research results [34]. The most frequently interrogated features are the extent of myocyte necrosis, the number of inflammatory foci per section, and the presence of fibrosis [31,33,35,36].

A comprehensive examination of the correlations between myocardial inflammation patterns and specific categories of causes of death holds promise for shedding further light on the role of myocarditis in mortality. Thus, our study aimed to delve into the spectrum of histological and immunohistochemical features of myocarditis within an adult autopsy population, encompassing forensic cases. Subsequently, we sought to establish connections between these histological findings and the documented causes of death as extracted from the autopsy reports.

## 2. Materials and Methods

This retrospective and observational study was conducted on 26 autopsy cases from a single centre, represented by The National Institute of Legal Medicine, over a period of 4 years (July 2018–December 2021). 

We interrogated the database of the Department of Histopathology for the word “myocarditis”. This was the sole inclusion criterion. However, we excluded cases with autolytic changes in order to avoid false reactions during immunohistochemistry testing (Figure 1).

In the selected cases, we gathered data from the histopathological reports concerning the circumstances of death, epidemiological details (age, sex), and microscopic characteristics of vital organs such as brain tissue, lungs, kidneys, and liver. All the collected data were entered into an anonymized Microsoft Excel worksheet (version 2021). Statistical analysis was conducted utilizing SPSS Statistics version 21 for Windows. All *p*-values were calculated using a two-tailed approach, and values below 0.05 were deemed statistically significant. The Mann–Whitney test was employed for comparing two groups when dealing with non-parametric data that did not adhere to a normal distribution. Nominal data were compared using a chi-squared test (χ^2^). The Spearman test was utilized as a statistical method to evaluate the strength and direction of the relationship between two variables, particularly in cases where the relationship between them was not assumed to be linear.

The histopathological reports and cardiac tissue blocks were retrieved and used for this study with institutional approval. For each case, sections from the left ventricular wall and/or the interventricular septum were available, ranging from one tissue block to three per case. Additional tissue sections were performed and utilized for haematoxylin-eosin staining and immunohistochemical analyses (CD3, CD163, and IL-6). Histological re-examination assessed variables such as fibrosis (Table 1), number of inflammatory foci, severity of inflammation (Table 2), and severity of necrosis (Table 3) using a grading system to be detailed in the following paragraphs and tables. 

Typically, inflammation in myocarditis manifests as focal, with foci dispersed throughout the ventricular and atrial walls. To quantify the degree of myocardial inflammation, we employed a graded classification system based on the percentage of medium-power fields (10× objective) containing inflammatory foci out of the total examined section (Table 2). Given that sections are predominantly sampled from the left ventricular wall or interventricular septum, their dimensions exhibit minimal variability within the case cohort. In 2010, an alternative system for grading inflammation was proposed [37]. The Kitulwatte scheme, which classifies inflammatory index, relies solely on the number of foci per slide, without considering their size. Consequently, a case exhibiting extensive myocarditis may be erroneously assigned a low index. Nevertheless, we incorporated this grading modality to facilitate comparison between the two scoring systems.

For the immunohistochemistry analysis, we utilized paraffin-embedded tissue blocks from our archives, which were subsequently sliced into thin sections (3–4 micrometres thick) using a microtome. These sections were then processed to remove the paraffin and rehydrate them before undergoing antigen retrieval. CD3 staining was performed to identify T lymphocytes, as it is a marker present in all subsets of normal T lymphocytes and aligns with the guidelines established by the European Society of Cardiology Working Group on Myocardial and Pericardial Diseases. The staining methods were conducted automatically using the Ventana Benchmark Ultra system and the 2GV6 Rabbit Monoclonal Antibody. We specifically assessed membranous reactions in small lymphocytes, quantifying the percentage of positive cells within the entire inflammatory focus. Lymphocytes located within blood vessels were disregarded and not included in the cell count. Macrophage infiltration was evaluated using CD163 instead of CD68, as the former marker showed minimal background staining. Immunostaining was conducted using the Leica Bond-III automated immunostainer and the 10D6 Mouse Monoclonal Antibody. Positive reactions were observed in both membranous and cytoplasmic compartments within the inflammatory foci. Macrophages found outside the regions of myocarditis were excluded from the analysis and regarded as tissue-resident macrophages. IL-6 expression was assessed using a manual protocol tailored for the anti-human antibody (NCL-L-IL-6 Leica Biosystems clone 10C12 Mouse Monoclonal Antibody). Initially, the slides underwent deparaffinization followed by a 20 min incubation at 97 °C. Subsequently, they were washed and exposed to peroxidase for 10 min, then incubated with a protein-blocking solution for an additional 5 min before being washed with PBS again. The subsequent steps were incubation with anti-IL-6 for 30 min, washing with PBS, and counterstaining with haematoxylin. Cytoplasmic positivity was noted in several immune cells within the myocarditis foci. As part of this particular manual protocol, colon mucosa served as the positive external control.

The immunohistochemical expressions were assessed quantitatively as the percentage of positively stained cells for each of the three markers, encompassing the entirety of the inflammatory infiltrate. Neutrophils were additionally included in this enumeration due to their distinctive morphological attributes, characterized by multilobed nuclei and amphophilic cytoplasm, given the absence of a routinely employed specific stain for their identification.

## 3. Results

### 3.1. Demographic Findings

The research encompassed the examination of 26 forensic cases, predominantly involving male individuals (73%, 19 out of 26), with an average age of 51.6 years (CI: [43.52, 59.71]) and a median age of 51.5 (range 18–83). Notably, the male participants in our sample exhibited a slightly higher age distribution compared to the females within our cohort (52.7 years versus 48.5 years). The circumstances surrounding these fatalities demonstrated considerable diversity, as delineated in Table 4. In addition to cases deemed forensic due to traumatic events, a number of autopsies were performed in response to malpractice claims or because the deaths occurred in public places under unknown circumstances. In the clinical files documenting the context prior to autopsies, sudden death and drug abuse were the most frequent set of conditions to be associated with myocarditis findings on histological examination (38%). 

Individuals with substance use disorders ranged in age from 18 to 44 years, with a mean age of 31.09 years (standard deviation 9.28; CI: [24.85; 37.33]) and a median of 30 years (range 18–44). The non-drug user group ranged in age from 18 to 89 years, with a mean age of 56.21 years (standard deviation 17.19; CI: [51.99; 60.44]) and a median of 57.50 years. The Mann–Whitney U test was applied to determine if there were differences between the group of substance users and the group of non-drug users regarding age. There were statistically significant differences in the age between individuals in the substance user group (mean rank 13.09) and those in the non-drug user group (mean rank 43.32), U = 78.000, Z = ‒4.150, *p* ≤ 0.001, underlining that the group of substance users is considerably younger than the non-drug-consuming group. A notable finding emerged regarding the incidence of pneumonia/bronchopneumonia among patients with substance use disorders. The data revealed a substantially higher prevalence within this group compared to non-drug-consuming individuals. Specifically, a significant majority of substance users, accounting for 72.73% of cases, exhibited histological features of pneumonia/bronchopneumonia. In contrast, only 34.85% of non-users were diagnosed with pneumonia/bronchopneumonia. Further analysis utilizing the chi-squared test unveiled a statistically significant relationship between pneumonia/bronchopneumonia and substance abuse (χ^2^ = 5.625, *p* = 0.018). Moreover, the calculated phi and Cramer’s V coefficients (0.270) indicated a direct yet weak association between these variables (*p* = 0.018).

### 3.2. Histological Examination

Histological analysis using routine haematoxylin-eosin staining predominantly showcased cardiac tissue slides exhibiting reduced inflammation levels, with 38.4% graded as mild (grade 1) and 34.6% as moderate (grade 2), while only 26% of cases displayed high-grade inflammation (Figure 1). In terms of inflammatory foci count, the average was 4.9 foci per section, ranging from 1 to 12. A Spearman correlation coefficient was calculated to assess the relationship between inflammation severity and the number of inflammatory foci per section in the studied cases. The results revealed a strong and direct correlation between inflammation severity and the number of inflammatory foci per section, with rs = 0.814, *p* ≤ 0.001. Thus, in the cases examined, inflammation severity increases as the number of inflammatory foci per section is higher.

The subset of individuals experiencing sudden death exhibited a lower degree of inflammation, with 83.3% of cases showing mild to moderate inflammation (grade 1–2), and an average of 2.8 foci per section.

Necrosis was identified in the haematoxylin-eosin stain in regions where the cardiac syncytium had been lost and replaced by inflammatory cells and necrotic debris. These areas are characterized by the absence of intact cardiac cells and the presence of cellular debris. In addition, we included nearby cardiomyocytes that had lost their striations and demonstrated a more intense eosinophilic cytoplasm, indicating early stages of cellular injury and degeneration. Hence, necrosis grading correlated with the inflammation pattern, with the majority (80.7%) showing low-grade necrosis (12 cases—grade 1, 9 cases—grade 2).

Slides with extensive necrosis also exhibited a predominantly neutrophilic infiltrate in the myocarditis foci (Figure 2). Additionally, among these five cases, two exhibited the presence of Aspergillus hyphae and one associated bacterial colony. The Mann–Whitney U test was applied to determine if there are differences between the group with microorganisms and the group without microorganisms regarding neutrophil levels. There are statistically significant differences in neutrophil percentage between individuals in the group with microbial colonies (mean rank 17.44) and those in the group without microbial colonies (mean rank 10.91), U = 32.500, Z = –2.075, *p* = 0.038. Thus, in cases with predominantly neutrophilic inflammatory infiltrate, the presence of intralesional pathogens is significantly more frequent.

In our study group, the assessment of fibrosis revealed that the majority of cases exhibited adventitial fibrosis (42%—grade 1, 11 out of 26) or insignificant fibrosis (27%—grade 0, 7 out of 26) (Figure 3). Among the remaining five cases with moderate to severe fibrosis, there were concomitant observations of arteriosclerosis and myocyte hypertrophy. There were no statistically significant correlations found between age or the circumstances of death in these cases.

### 3.3. Immunohistochemical Results

In our cohort, the predominant inflammatory cell types identified were macrophages and neutrophils, both exhibiting similar percentages across the entire group. Specifically, macrophages demonstrated a mean prevalence of 31.92% and a median of 30% (range 8–70), while neutrophils exhibited a mean prevalence of 32.2% and a median of 42.5% (range 0–80). In contrast, lymphocytes constituted a minority within the myocarditis foci of our forensic cases.

To elucidate the inflammation patterns and cellular dynamics within the studied group, we conducted comprehensive statistical analyses. These analyses were aimed at identifying and describing correlations between the presence of various inflammatory cells and the corresponding clinical contexts. By employing such methodologies, we were able to provide a more detailed and nuanced understanding of the inflammatory processes underlying myocarditis in our cohort.

In our series of cases, protein CD3 was scarcely present in the inflammatory infiltrate, with an average value of 15.19% in the inflammatory infiltrate (standard deviation 22.15; CI: [6.24; 24.14]), and a median value of 4% (range 1–90). Although the differences between genders are not statistically significant, it was observed that males tended to have slightly higher CD3 values compared to females (*p* = 0.334), with males having an average value of 16.84% (standard deviation 23.53; CI: [5.50; 28.18]) and a median of 5% (range 1–90), while females had an average value of 10.71% (standard deviation 18.74; CI: [−6.62; 28.05]) and a median of 1% (range 1–50). The Spearman correlation coefficient was calculated to assess the relationship between CD3 and the severity of inflammation measured in the patient cohort. There is a moderate inverse correlation between the CD3 values and the severity of inflammation, rs(24) = –0.476, *p* = 0.014. Thus, in the cases studied, as the severity of inflammation increases, the CD3 protein value decreases. The same moderate inverse correlation was found between the CD3 values and the severity of necrosis, rs(24) = –0.506, *p* = 0.008. In the same manner, we discovered a monotonic relationship between CD3 and neutrophil levels measured in the patient cohort. There is a very strong inverse correlation between CD3 level and neutrophil level, rs(23) = –0.875, *p* ≤ 0.001. Thus, in the studied cases, as the neutrophil level increases, the CD3 protein value decreases, indicating a different mechanism of the immune response. Without marking statistically significant associations, it was noted that the CD3 level was higher in patients who experienced sudden death compared to those who died under other circumstances (mean CD3 value: 31.40% vs. 11.33%). (*p* = 0.278.) Moreover, the CD3 level is lower where inflammatory foci are numerous (*p* = 0.07), thus emphasizing the discreet nature of lymphocytic myocarditis (Figure 4).

Macrophage presence within the inflammatory infiltrate was assessed using CD163, selected for its specificity and absence of typical background staining associated with CD68. The proportion ranged from 10% to 70% in cardiac inflammation, with a mean of 31.92% (SD 15.56; CI: [25.64; 38.21]), and a median of 30% (range 10–70) (Figure 4). Although gender differences were not statistically significant, it was noted that CD163 levels tended to be slightly higher in women compared to men (*p* = 0.279). Specifically, the mean CD163 value for females was 37.86% (SD 17.76; CI: [21.43; 54.28]), whereas for males, it was 29.74% (SD 14.57; CI: [22.71; 36.76]). According to the results of the Mann–Whitney U test, employed to determine if there are differences between the group experiencing sudden death and the group deceased under other circumstances, there are statistically significant differences in CD163 values among patients in the former group (mean rank 7.40) compared to the non-sudden death group (mean rank 14.95), U = 22.000, Z = –2.014, *p* = 0.049. Hence, sudden death is more inclined to be linked with myocardial inflammation characterized by a reduced presence of macrophages. Although statistically significant correlations were not recorded, it is noteworthy that patients with pneumonia/bronchopneumonia-associated lesions recorded higher CD163 values compared to cases without this diagnosis (mean value 38.75% vs. 28.89%) (*p* = 0.196). A Pearson correlation coefficient was computed to assess the relationship between CD163 and neutrophil levels measured in the patient cohort. The graphical representation indicates a monotonic relationship. There is a moderate and inverse correlation showing that as neutrophil levels increase, CD163 protein values decrease, rs(22) = –0.470, *p* = 0.020. 

For IL-6 expression in the studied cohort, the values ranged from 0% to 15%. The mean IL-6 value was 3.84% (standard deviation 4.34; CI: [2.05; 5.63]), with a median value of 1% (range 0–15), without significant differences between the genders: (mean(F) = 5.5%, mean(M) = 3.32%). 

A Spearman correlation coefficient was calculated to assess the relationship between IL-6 and the severity of inflammation measured in the patient cohort. There is a moderate and direct correlation between IL-6 values and the severity of inflammation, rs(23) = 0.439, *p* = 0.028, indicating a monotonic relationship between the two variables, such that as inflammation severity increases, IL-6 protein value also increases. The same test applied to the relationship between IL-6 expression and the severity of necrosis identified a relatively strong direct correlation between IL-6 values and the severity of necrosis, rs(23) = 0.547, *p* = 0.005. Furthermore, the same monotonic relationship was observed between the level of IL-6 and the number of inflammatory foci/section measured in the patient cohort, indicating a moderate direct correlation between IL-6 level and the number of inflammatory foci, rs(23) = 0.402, *p* = 0.047. When compared to the expression of CD3, the Spearman correlation coefficient indicates a monotonic relationship between these two variables, indicating a moderate inverse correlation between CD3 values and IL-6, rs(23) = –0.548, *p* = 0.005 (Figure 5).

## 4. Discussion

In our cohort of forensic autopsy cases, myocarditis was observed microscopically within a highly variable context. The circumstances surrounding death primarily implicated patients with a background of altered immune response due to chronic conditions or prolonged intensive-care hospitalization in a traumatic context. While this presentation diverges from the typical scenario for myocarditis diagnosis, our series of patients also exhibited a predominantly male population (73%). The majority fell within the fifth to sixth decade of life, slightly older than the typically recognized demographic of young adults who commonly experience myocarditis [5,6]. Notably, the youngest subset in our study (with a mean age of 31) consisted of individuals with a documented history of drug usage, predisposing them to severe pulmonary infections and neutrophilic myocarditis compared to non-drug abusers. Sudden death constituted another prevalent category in our myocarditis cases. The average and median ages of individuals in this category were consistent with the overall population in the study. However, histological analysis revealed that myocardial inflammation in this subgroup was generally mild, with fewer foci compared to other cases. These findings align with recent studies on sudden death, which predominantly identified lymphocytic infiltrates in cardiac tissue samples obtained during autopsies. The results indicate that lymphocytic myocarditis accounted for more than half of the cases reviewed, with neutrophilic inflammation present in nearly 20% of the cases [38,39].

The inflammatory infiltrate was quantitatively assessed using a grading system and an adjusted counting method previously proposed [33]. It has been demonstrated that assessing the extent of inflammation and cellular damage in myocardial tissue enhances the ability to accurately predict a patient’s risk of adverse disease progression and to adjust the treatment accordingly [40,41]. Despite the absence of standardized and validated methods for assessing these parameters, pathologists are encouraged to report the percentage or proportion of myocardial necrosis and the extent of inflammation observed in the slides they examine. The Kitulwatte scheme, which was suggested to evaluate the extent of inflammation in myocarditis by counting the foci/slide, aligns with our approach, which emphasizes the measurement of the size of inflammatory foci per section (*p* ≤ 0.001) [37].

To enhance the accuracy of histological examination, immunohistochemistry was employed to better delineate the population of inflammatory cells and their distribution within cardiac tissue inflammation. According to the current literature, the most prevalent cells identified are T lymphocytes, characterized by CD3 expression, and macrophages, identified through CD163 staining [42,43]. Additionally, neutrophils were included in this study due to their significance within the post-mortem context of myocarditis. A comprehensive review of the forensic pathology literature indicates that myocarditis cases with neutrophil- and macrophage-rich polymorphous infiltrates suggest an infectious, primarily bacterial cause [44]. These observations are supported by the physiological innate immune response, which employs phagocytosis, cytokine release, and complement activation to eliminate these microorganisms, even in the absence of a specific antibody-mediated response. In our study, cases with severe inflammation, predominantly neutrophilic, and necrosis were frequently associated with bacterial or fungal myocarditis, confirmed by tissue section analysis or cardiac blood testing (*p* = 0.038).

In the presented retrospective research, the majority of cases (61.5%) exhibited a low CD3 count (less than 7% in the inflammatory infiltrate), suggesting that CD3 should not be considered significant in autopsy cases, as most post-mortem diagnoses may reflect complications from sepsis, endocarditis, or altered immune status. Furthermore, studies on endomyocardial biopsies indicate that a pan-leukocyte marker is more sensitive for identifying myocarditis, according to the guideline-recommended cut-off (14 leukocytes per mm^2^) [45]. However, this approach is more applicable to small samples with limited tissue availability [46].

In instances of sudden death, detection of CD3 in cardiac muscle may support the diagnosis of an insidious or unknown cardiomyopathy, often associated with fatal arrhythmias [47]. It is likely that recommendation for immunohistochemical testing of CD3 should be considered only under specific conditions, after excluding a clinical context favourable to sepsis or histologically excluding myocarditis with neutrophils, considering the inverse relationship between lymphocytes and neutrophils in cardiac inflammation.

The immunohistochemical analysis targeting CD163+ macrophages revealed their representation at approximately one-third of the inflammatory cells in myocarditis. This proportion tends to exhibit an inverse correlation with the abundance of neutrophils, suggesting a dynamic interplay between these cell types in myocardial inflammation. Similar significant observations were reported in prospective studies on endomyocardial biopsies, which found that a high proportion of CD163 macrophages serves as a predictive factor for poor outcomes (*p* = 0.004) [18]. Furthermore, our results highlight that a reduced macrophage count was notably observed in patients who succumbed to sudden death. Comparably, murine models with macrophage depletion also resulted in sudden death [48]. Altogether, these results emphasize the diagnostic utility of illustrating these cells using markers such as CD163 or CD68 in the evaluation of myocarditis. Additionally, this information can influence clinical practice, as changes in plasma levels of CD163 have been shown to precede adverse cardiovascular events in patients with chronic heart failure [49,50].

In recent years, numerous investigations have delved into the assessment of various proteins implicated in inflammatory pathologies, aiming to uncover novel therapeutic strategies employing immunomodulatory drugs [20,23,24,51]. Notably, IL-6, a key pro-inflammatory cytokine, has been identified as significantly upregulated in cardiac tissue during severe manifestations of myocarditis, alongside elevated serum levels. In our investigation, utilizing immunohistochemical analysis, we discerned a moderate positive correlation between IL-6 expression levels and the severity of inflammation (*p* = 0.028), the necrosis severity (*p* = 0.005), and the number of inflammatory foci per section (*p* = 0.047), indicating a consistent linkage where heightened IL-6 levels align with increased severity of these pathological indicators. Furthermore, we observed a moderate inverse correlation between IL-6 expression and CD3 levels (*p* = 0.005), suggesting a relationship wherein reduced IL-6 expression associates with a predominance of lymphocytic myocarditis. These findings, consistent with prior literature, underscore the potential of myocardial IL-6 as a discerning marker of inflammation severity and necrosis. However, a limitation of this research is the absence of serum IL-6 level assessment, which precluded evaluation of the relationship between circulating IL-6 and myocardial involvement. Exploring inflammatory chemokines within myocarditis foci could elucidate their impact in particular clinical contexts.

## 5. Conclusions

Myocardial inflammation in forensic settings exhibits distinct characteristics compared to myocarditis identified through endomyocardial biopsies, with notable correlations to sudden death and drug abuse complications. These findings underscore the need for a unique immunohistochemical approach, emphasizing stains such as CD163 and IL-6, which correlate with inflammation and necrosis severity and may serve as biomarkers, potential therapeutic targets, or prognostic factors.

The study’s methodology is highly reproducible and suitable for daily practice. The use of stable and validated antibodies, such as CD3 and CD163, ensures consistent results. Although IL-6 is a novel immunohistochemistry marker, our detailed description of the manual process in the Materials and Methods Section facilitates easy replication. Additionally, the percentage scoring system for assessing positive reactions is straightforward, further enhancing the method’s applicability in routine laboratory settings.

## Data Availability

The data presented in this study are available on request from the corresponding author due to (legal reasons).

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
