# Peer review of "Novel Immunohistochemical and Morphological Approaches in a Retrospective Study of Post-Mortem Myocarditis"

_medicina, 2024, doi:10.3390/medicina60081312_

Round 1

Reviewer 1 Report

Comments and Suggestions for Authors

In this study Neagu et al have studied the role of IH analysis in myocarditis of autopsy cases.

The way the results are presented now makes it hard to understand putative myocarditis. What is the predominant cell type?

I have the following comments:

1) it still would be interesting to analyse also the Pan-macrophage marker CD68, as you miss macrophages using CD163 only.  The same is true for the Leukocyte marker CD45, that has been described to be an important marker to diagnose lymphocytic myocarditis in autopsy cases. The same is true for the neutrophilic granulocyte marker MPO.

2) Table 1 have some doubt about the fibrosis grading system. More appropriate would be: adventitial fibrosis, interstitial fibrosis and/or replacement fibrosis.

In case authors want to quantify fibrosis, you can quantify this in the slide.

3) Table 3: HE staining is used for necrosis: this will underestimate real necrosis: PTAH staining or C3d/C4d staining is more reliable.

4) I am missing Table 4 in the manuscript.

5) Results 3.1: the statement: Sudden death and drug abuse were the most frequent set of conditions to be associated with myocarditis findings on histological examination: this is not clear for me. What do you mean by this?

6) Figure 2: necrosis is not clear: use arrow to make your point.

7) Quantification of CD68, CD45, MPO, CD3 should be performed: then you know what we are looking at. What is the predominant cell type?. Now this is not clear. This should also be depicted in graphs.  

8) Il-6 scoring is not clear for me. Theoretically this can be secreted and could be positive in blood vessels, next to inflammatory cells. This should also be depicted in graphs.

Author Response

          Comment:  The way the results are presented now makes it hard to understand putative myocarditis. What is the predominant cell type?

           Response: We appreciate your feedback regarding the clarity of our results. In our cohort, the predominant inflammatory cell types were macrophages and neutrophils, with similar percentages across the entire group (the mean for macrophages was 31.8%, and for neutrophils, it was 32.2%). We have added this information to the manuscript and made adjustments to the text to enhance the clarity of our findings (page 10, 1st paragraph in section 3.3-Immunohistochemistry results, lines 341-352).

Response to the following comments:

1) it still would be interesting to analyse also the Pan-macrophage marker CD68, as you miss macrophages using CD163 only.  The same is true for the Leukocyte marker CD45, that has been described to be an important marker to diagnose lymphocytic myocarditis in autopsy cases. The same is true for the neutrophilic granulocyte marker MPO.

Response 1:

          Thank you for your insightful remarks. In our study, we focused on a set of antibodies aimed at providing a beneficial perspective on myocarditis within a different context than the classical one. We chose CD163 over CD68 due to its greater specificity and clearer staining pattern, as CD68 can also highlight fibroblasts, certain lymphoid cells, and typically shows background staining.

            Regarding CD45, we opted not to include this immunostain because it does not differentiate between inflammatory cells and does not provide additional information. Unfortunately, MPO is unavailable in our laboratory, and we identified granulocytes based on their distinctive nucleus and cytoplasm morphology. In summary, our goal was to present new information using a restricted set of antibodies that can add valuable insights and correlations to the clinical context.

2) Table 1 have some doubt about the fibrosis grading system. More appropriate would be: adventitial fibrosis, interstitial fibrosis and/or replacement fibrosis.

In case authors want to quantify fibrosis, you can quantify this in the slide.

Response 2:   Thank you for your valuable comment. We have clarified and added this information to both the table and the figure legend (page 5-Table 1; page 10-Figure 3).

3) Table 3: HE staining is used for necrosis: this will underestimate real necrosis: PTAH staining or C3d/C4d staining is more reliable.

Response 3:  We appreciate your insightful comments regarding the sensitivity of PTAH staining and complement activation markers (C3d/C4d) for detecting myocardial injury. Indeed, these methods offer greater sensitivity, particularly in cases where HE staining results are inconclusive, such as those involving a low count of inflammatory cells or ischemic injuries. For this particular research, however, we opted for HE staining due to its simplicity and widespread availability in all laboratories, which allows for a more accessible evaluation of myocardial damage.

4) I am missing Table 4 in the manuscript.

          Response 4:  We respect your careful review and apologize for the oversight. Table 4 was inadvertently omitted from the final version of the submitted manuscript. We have now corrected the text accordingly (page 7).

5) Results 3.1: the statement: Sudden death and drug abuse were the most frequent set of conditions to be associated with myocarditis findings on histological examination: this is not clear for me. What do you mean by this?

          Response 5:  We appreciate your pertinent question and thank you for the opportunity to clarify this statement. In Table 4, we listed all the circumstances of death observed in our cohort, referring to the known context prior to conducting the autopsies. The majority of cases identified as myocarditis in our study shared these two circumstances, as noted in the clinical files. We have revised and expanded the explanations in the last paragraph of page 6 to enhance clarity (lines 252-258).

6) Figure 2: necrosis is not clear: use arrow to make your point.

           Response 6: We appreciate your constructive feedback. In the four images presented in Figure 2, necrosis is depicted by the entire area devoid of cardiac cells, characterized by necrotic debris and inflammatory cells. Additionally, we included nearby cardiomyocytes that have lost their striations and exhibit a more intense eosinophilic cytoplasm. Due to these factors, it is challenging to effectively highlight necrosis with arrows. Nevertheless, we have added clarifying information in the text to enhance understanding (page 9, 2nd paragraph, lines 302-309).

7) Quantification of CD68, CD45, MPO, CD3 should be performed: then you know what we are looking at. What is the predominant cell type? Now this is not clear. This should also be depicted in graphs.  

           Response 7: Incorporating markers such as CD45 and MPO would indeed provide a more detailed depiction of the cellular inflammation. We will strive to integrate these markers into our laboratory's routine practice to enhance the thoroughness of our analyses. However, we are confident that the panel employed in this study is suitable to highlight certain specificities relevant to the forensic context of myocarditis. The selected markers allow us to delineate the inflammatory profile effectively within the constraints of our current methodology. Moreover, we hope that clinicians will consider investigating both serological and tissue expressions of inflammatory markers such as CD163 and IL6, as these can offer additional information into the inflammatory processes. We acknowledge the importance of depicting this data in graphical form for clarity and will aim to include such representations in future studies.

8) Il-6 scoring is not clear for me. Theoretically this can be secreted and could be positive in blood vessels, next to inflammatory cells. This should also be depicted in graphs.

       Response 8:     We acknowledge that IL-6 can indeed be secreted by various cell types and may be present in blood vessels, next to inflammation sites. Therefore, we focused our analysis on the positive expression of IL-6 specifically within myocarditis foci, counting only positive lymphocytes, macrophages, and neutrophils. Intravascular inflammatory cells or any background staining were not included in our histological and immunohistochemical analyses.

Although depicting the positive cells in our images can clarify this aspect, we are confident that the positive cells in our images accurately reflect the cellular density in specific myocarditis cases. For this reason, we decided to keep the images clearer by refraining from adding arrows.

Reviewer 2 Report

Comments and Suggestions for Authors

Background: Macrophages are known to participate in various biological functions, playing a dual role in the process of myocardial regeneration. Also, macrophages, in addition to differences in the functions they perform, have the property of plasticity, which allows them to change their phenotype and functions in response to signals from the microenvironment. Molecular biomarkers of monocytes/macrophages known to date have demonstrated broad diagnostic capabilities. The diversity of monocyte/macrophage phenotypes requires further study and standardization. Strengths of the study: the data obtained during the study on macrophage infiltration of the myocardium are both fundamental and applied in nature, which is of interest to both scientists and practical healthcare. The data obtained by the authors may contribute to the development of new pharmacological approaches in certain patient groups. Weaknesses of the study: 1) small number of observations; 2) Table 4 is not presented; 3) I propose to specify whether there were cases of death of patients from myocardial infarction, since for such patients there are features of the inflammatory reaction in the myocardium, associated, among other things, with the time from the development of myocardial infarction to the moment of death; 4) since the article deals with myocarditis, I propose to clarify whether the expression of antigens of cardiotropic viruses in the myocardium was determined; 5) I propose to clearly indicate which main macrophage markers were considered as a marker of M1 macrophages, and which ones were considered as a marker of M2 macrophages.

Author Response

Weaknesses of the study: 1) small number of observations;

Response 1: We acknowledge that our cohort consists of only 26 cases with confirmed myocardial inflammation, which may fall short of the statistical power required for general population. However, we believe that our study offers significant value by focusing on specific contexts and providing in-depth insights into a particular form of myocardial impairment. Moreover, our inclusion of immunohistochemistry tests for markers such as CD163 and IL-6, in addition to the classical CD3 required for the morphological diagnosis of myocarditis, enhances the profiling of these proteins at the site of inflammation.

In order to highlight the relevance of our observations, we have revised the discussion section. Please find the updated version providing a more comprehensive context for our results in Section 4, Page 13: Paragraphs 1-2 (lines 426-441), Paragraphs 3-4 (lines 446-463), and the last paragraph (lines 474-485, Page 14).

While we recognize the limitation in the number of observations, we are confident that our findings contribute meaningful data to the field and that subsequent studies will build upon our clinico-pathological correlations and further investigate the complexities of myocardial inflammation.

2) Table 4 is not presented;

Response 2: We respect your careful review and apologize for the oversight. Table 4 was inadvertently omitted from the final version of the submitted manuscript. We have now corrected the text accordingly (page 7).

3) I propose to specify whether there were cases of death of patients from myocardial infarction, since for such patients there are features of the inflammatory reaction in the myocardium, associated, among other things, with the time from the development of myocardial infarction to the moment of death;

Response 3: We have included in our research only cases with histologically confirmed myocarditis, which typically exhibits a different pattern of distribution compared to myocardial infarction. In myocarditis, the inflammation is patchy and unevenly distributed, consisting of a mixture of inflammatory cells. In contrast, myocardial infarction is characterized by diffuse necrosis that follows the vascular territory. To ensure the accuracy of our study, we thoroughly reviewed all slides and excluded cases with equivocal morphology, such as diffuse ischemic changes, autolysis or severe coronary atherosclerosis.

4) since the article deals with myocarditis, I propose to clarify whether the expression of antigens of cardiotropic viruses in the myocardium was determined;

Response 4: We appreciate your insightful comment on this matter. While our current autopsy protocol does not include the detection of viral antigens in cardiac tissue due to specific logistical constraints, we acknowledge that incorporating such testing would have enriched our study. Although the primary aim of our research was not to elucidate the aetiology of myocarditis, given the complex clinical contexts of our cases, we recognize the importance of viral load testing, especially in lymphocytic myocarditis.  Moving forward, we hope to incorporate this aspect into future studies to provide a more comprehensive analysis of the aetiology of myocarditis.

5) I propose to clearly indicate which main macrophage markers were considered as a marker of M1 macrophages, and which ones were considered as a marker of M2 macrophages.

Response 5: We appreciate your remark regarding the distinction between macrophage subtypes. While this would indeed be a valuable differentiation, it is important to note that classical immunohistochemistry using markers such as CD163 or CD68 does not allow for a clear distinction between M1 and M2 macrophages. In our study, we selected CD163 due to its greater specificity for macrophages and the clearer background it provides. However, it is worth mentioning that CD163 is expressed on both M1 and M2 macrophages, and therefore, does not enable differentiation between these subsets.

Reviewer 3 Report

Comments and Suggestions for Authors

This is an exciting research paper.

However, a few suggestions are placed to further improve the manuscript.

Introduction: Nicely written.

Comment 1: Introduction is well written. However, it is too long and needs to focus on what lacune will be filled by the present study. The available literature must be smartly curtailed so that the size is reasonable.

Method:

Comment 2: For this retrospective study, a flow diagram of patient search methodology should be added to make a better reading experience and better flow.

Results:

Comment 3: Please add range with median values (age, etc).

Comment 4: For identifications of T cell infiltrates, why only CD3 antibodies were chosen, and why not a combinations of antibodies (details of it may also be included in method section).

Discussion:

Comment 5:  the discussion needs to be written with respect to the findings of the study and should be compared with the available literature with appropriate references.

Comment 6:  The conclusion must be crisply written in one to two lines.  

Reference: insufficient

Table and Figure: looks good

Author Response

Comment 1: Introduction is well written. However, it is too long and needs to focus on what lacune will be filled by the present study. The available literature must be smartly curtailed so that the size is reasonable.

Response 1: Thank you for your positive feedback. In accordance with your suggestion, we have revised the introduction to reduce its length, retaining only the most relevant information for our research. This revision ensures a concise portrayal of the current state of knowledge concerning the contributions of CD163 and ILs as biomarkers and their tissue expression. The modifications can be found on page 2, 2nd paragraph (lines 71-75), 3rd paragraph (lines 87-90 and 97-99), and page 3, 2nd and 3rd paragraphs. Overall, we have reduced the introduction by 41 lines to better suit the background of our study.

Method:

Comment 2: For this retrospective study, a flow diagram of patient search methodology should be added to make a better reading experience and better flow.

Response 2: Thank you for your valuable suggestion. We have created and included a flow diagram of the patient search methodology to enhance the reading experience and provide a clearer flow of the retrospective study. We believe this addition will significantly improve the understanding of our methodology (refer to page 4, section 2).

Results:

Comment 3: Please add range with median values (age, etc).

Response 3: Thank you for highlighting this. We have added the range alongside each mention of a median value in the results section, specifically in Section 3.1 on page 6 (first paragraph, line 226), page 7 (line 239), page 10 (lines 331, 334, 335, and 354), and page 11 (line 380).

Comment 4: For identifications of T cell infiltrates, why only CD3 antibodies were chosen, and why not a combination of antibodies (details of it may also be included in method section).

Response 4: Thank you for your insightful comment. We chose CD3 as the sole marker for identifying lymphocytes because it is present in all subsets of normal T lymphocytes and aligns with the guidelines set forth by the European Society of Cardiology Working Group on Myocardial and Pericardial Diseases. Additionally, given the limited number of cases with a significant lymphocytic infiltrate, we did not deem it necessary to include additional immunostains for these types of inflammatory cells. We will include these details in the methods section for clarity.

Discussion:

Comment 5:  the discussion needs to be written with respect to the findings of the study and should be compared with the available literature with appropriate references.

Response 5: Thank you for your valuable feedback. In response to your suggestion, we have revised the discussion section to align it more closely with the findings of our study. We have also incorporated relevant and recent literature with similar findings to provide a more comprehensive context for our results. Please find the modifications in Section 4, Page 13: Paragraphs 1-2 (lines 426-441), Paragraphs 3-4 (lines 446-463), and the last paragraph (lines 474-485, Page 14).

Comment 6:  The conclusion must be crisply written in one to two lines.  

Response 6: Thank you for pointing this out. We have revised the conclusion to ensure it is concise and highlights the main ideas of our study. Please refer to page 14, lines 505-509.

Reference: insufficient

Response 7: Thank you for your feedback. We have added additional references to strengthen our manuscript and the discussion section. Please find the updated reference list from numbers 39 to 52 on page 17.

Round 2

Reviewer 1 Report

Comments and Suggestions for Authors

Authors did not respond properly to most of my comments

Author Response

We greatly appreciate the time and effort you have taken to review our manuscript and provide insightful feedback. Your comments have been invaluable in guiding us to enhance the clarity and robustness of our study. We understand that your comments were primarily regarding our chosen immunohistochemistry panel.

We acknowledge your concerns regarding the restricted nature of our immunohistochemistry panel. However, we would like to emphasize that our chosen markers, specifically for macrophages (CD163) and T lymphocytes (CD3), are highly specific and easily reproducible in clinical settings. These markers provide a rapid and reliable method for addressing myocardial inflammation, which is crucial for clinicians.

Furthermore, we have incorporated IL-6 into our panel, using a percentage scoring system at the site of inflammation. This addition is particularly valuable as it may help explain more aggressive cases of myocarditis. We believe that IL-6, as a novel marker, holds significant potential to become an essential component in the evaluation of endomyocardial biopsies, especially in clinically relevant cases.

We hope this addresses your concerns and demonstrates our commitment to improving the diagnostic tools available for myocardial inflammation. Thank you for your understanding.

Reviewer 2 Report

Comments and Suggestions for Authors

the reviewer is satisfied with the answers. The manuscript can be published

Author Response

We are delighted to receive your positive feedback. Your thorough review and constructive comments  significantly contributed to the improvement of our manuscript.

Thank you for your recommendation for publication. 

Reviewer 3 Report

Comments and Suggestions for Authors

well modified

Author Response

We are grateful for your positive feedback. Your thorough review and valuable comments have been instrumental in refining our manuscript.